# Rapid and Visual Detection of Volatile Amines Based on Their Gas–Solid Reaction with Tetrachloro-p-Benzoquinone

**DOI:** 10.3390/molecules29081818

**Published:** 2024-04-17

**Authors:** Yue-Xiang Sun, Zi-Jian Yan, Wan-Xia Liu, Xiao-Ming Chen, Man-Hua Ding, Lin-Li Tang, Fei Zeng

**Affiliations:** Department of Biology and Chemistry, Hunan University of Science and Engineering, Yongzhou 415199, China; syx18216222607@163.com (Y.-X.S.); y13908492281@yeah.net (Z.-J.Y.); 19151935043@163.com (W.-X.L.); chenxiaoming3012@163.com (X.-M.C.); dingxiaolove@163.com (M.-H.D.)

**Keywords:** amines, gas–solid reactions, tetrachloro-p-benzoquinone, detection, color change

## Abstract

The detection of volatile amines is necessary due to the serious toxicity hazards they pose to human skin, respiratory systems, and nervous systems. However, traditional amines detection methods require bulky equipment, high costs, and complex measurements. Herein, we report a new simple, rapid, convenient, and visual method for the detection of volatile amines based on the gas–solid reactions of tetrachloro-p-benzoquinone (TCBQ) and volatile amines. The gas–solid reactions of TCBQ with a variety of volatile amines showed a visually distinct color in a time-dependent manner. Moreover, TCBQ can be easily fabricated into simple and flexible rapid test strips for detecting and distinguishing *n*-propylamine from other volatile amines, including ethylamine, *n*-butyamine, *n*-pentamine, *n*-butyamine and dimethylamine, in less than 3 s without any equipment assistance.

## 1. Introduction

Organic amines are important intermediates in the chemical, pharmaceutical, and food industries [1,2,3,4,5]. It is also considered one of the sources of grievous social and health problems due to the fact that organic amines have acute or delayed toxicities to human skin, respiratory systems, nervous systems, urinary systems, and hematopoietic systems [6,7]. In recent years, with the increasing demand for the environment and food safety, various methods such as gas chromatography [8,9,10], electrochemistry [11], optical spectrometries [12,13,14,15], and fluorescent methods [16,17,18,19,20,21,22] for the detection of amines have been developed. However, all of the above methods have the drawbacks of bulky equipment size, high costs, and complexity in measurement. In addition, the determination and discrimination of aliphatic amines still remain a challenge. Therefore, it is very urgent but important to develop a simple, rapid, convenient, and visual method for the detection of aliphatic amines.

Tetrachloro-p-benzoquinone (TCBQ) is a perhalogenated quinone compound that can act as a mild oxidant and has been widely used in the fields of pharmaceuticals, the chemical industry, and pesticides. In addition, the nucleophilic substitution of TCBQ by nucleophilic substances, such as aromatic and aliphatic amines or proteins, has also been widely studied [23,24,25,26,27]. Primary and secondary aliphatic amines are known to form the disubstituted products of TCBQ, and the color of the reaction solution changes from yellow to reddish brown. However, most of the reactions between aliphatic amines and TCBQ are carried out in the solution station, and the gas–solid reaction between volatile amines and TCBQ has rarely been reported. Gas–solid reactions have the advantage of a faster reaction speed, and the reaction can occur in a shorter time. Moreover, the reaction can occur at room temperature, thus saving the energy consumption of the reaction. We suspected that TCBQ could be used as the sensor for the simple, rapid, convenient, and visual method for the detection of volatile amines if volatile amines and TCBQ can react in the gas–solid station and result in clear color changes.

Herein, we report a simple, rapid, convenient, and visual approach for the detection of volatile amines based on the gas–solid reaction between volatile amines and TCBQ (Figure 1). The gas–solid reaction between volatile amines and TCBQ was first verified by powder X-ray diffraction and single crystal X-ray analysis, as well as the color change. Notably, gas sensors based on the TCBQ showed an excellent volatile amine sensing performance with the response rate in the order of *n*-propylamine > dimethylamine > *n*-butylamine > ethylamine > *n*-amylamine > *n*-hexylamine. Moreover, a test strip based on TCBQ was prepared, and *n*-propylamine could be distinguished from ethylamine, *n*-butyamine, *n*-pentamine, *n*-butyamine, and dimethylamine in less than 3 s without any equipment assistance. Our research provides a new strategy for the rapid, convenient, and visual detection of volatile amines.

## 2. Results and Discussion

Initially, we tested the possibility of a gas–solid reaction between volatile amines and TCBQ. As shown in Figure 2b, after an open vial (4 mL) containing 25 mg of TCBQ was placed into a sealed vial (20 mL) containing 1 mL of *n*-propylamine, the color of TCBQ changed from yellow to brown within 30 s, indicating that a reaction between TCBQ and *n*-propylamine occurred. The ^1^H NMR spectrum of the formed brown solid showed the signals of 2,5-dichloro-3,6-bis(propylamino)cyclohexa-2,5-diene-1,4-dione, indicating that the nucleophilic substitution of the gas–solid reaction of *n*-propylamine with TCBQ successfully occurred (Appendix A). The resulting product was brown, probably due to the push–pull interaction between the electron-donating groups of amine and electron-withdrawing groups of carbonyls. Luckily, a single red-brown crystal suitable for X-ray analysis was obtained by the slow evaporation of the formed brown solid solution in dichloromethane (CCDC: 2345116), providing unambiguous evidence for the formation of 2,5-dichloro-3,6-bis(propylamino)cyclohexa-2,5-diene-1,4-dione (Figure 3b). In addition, the powder X-ray diffraction (PXRD) patterns of TCBQ after exposure to *n*-propylamine showed different sharp peaks with TCBQ (Figure 3f). All the above results indicate that TCBQ can form disubstituted products when exposed to *n*-propylamine vapor (Figure 2a). Similar to the case of *n*-propylamine vapor, we found that TCBQ could also form disubstituted products with ethylamine, *n*-butylamine, *n*-amylamine, *n*-hexylamine, and dimethylamine vapor, respectively. Interestingly, the various disubstituted products of TCBQ obtained by exposing TCBQ to different vapor amines were visually distinct in color in a time-dependent manner. As shown in Figure 2c, the color of TCBQ turned brown the fastest with *n*-propylamine vapor (within 30 s), and overall colorizing rates were in the order of *n*-propylamine > dimethylamine > *n*-butylamine > ethylamine > *n*-amylamine > *n*-hexylamine. The color of TCBQ changed to black after exposure to dimethylamine vapor for 360 s (Appendix A), while amine vapor changed the color of TCBQ to brown within 8–120 min (Appendix A).

The ^1^H NMR spectral of the formed color was solid, showing the signals of corresponding disubstituted products of TCBQ with tested amine vapor and providing further evidence for the gas–solid reaction between TCBQ and amine vapor (Appendix A). In addition, we also obtained the single crystals of the corresponding disubstituted products of TCBQ with ethylamine, *n*-butylamine, *n*-amylamine, and *n*-hexylamine, respectively. As shown in Figure 3, all of these crystals belong to the same triclinic crystal system and P-1 space groups, except for the disubstituted products of TCBQ by *n*-butylamine, which has the monoclinic crystal system and P2_1_/*n* space groups. Moreover, N-H···O hydrogen bonding interactions with distances of 2.131–2.258 Å were observed in these crystals, resulting in the formation of a liner supramolecular array architecture of corresponding disubstituted products of TCBQ in the solid states. The powder X-ray diffraction (PXRD) patterns of TCBQ after exposure to the amine vapor showed different sharp peaks with TCBQ, further supporting the formation of new compounds (Figure 3f).

Given the fact that TCBQ is time-dependent on the color change when exposed to different types of amine vapor, we wondered whether this feature could be exploited to distinguish different amines. Considering the feasibility of its practical application as well as portability, we then prepared TCBQ-based rapid test papers. TBCQ solution was first obtained by dissolving 100 mg of TBCQ in 10 mL of CH_2_Cl_2_, which was then dropped on the filter paper and dried at room temperature for 2 h. After the prepared test paper was exposed to different types of saturated amine vapor for 3 s, the color change was recorded. As shown in Figure 4, the significant color change can be visualized by the naked eye when the test paper was exposed to *n*-propylamine vapor, while other amine vapors showed no or little color change, suggesting that TCBQ possesses practical applications in detecting and distinguishing *n*-propylamine from ethylamine, *n*-butyamine, *n*-pentamine, *n*-butyamine, and dimethylamine in less than 3 s. To the best of our knowledge, this is the first TCBQ-loaded test strip not only for detection but also for the identification of *n*-propylamine from other amines, which presents promising practical applications for simplicity, ease, and speed of detection.

## 3. Materials and Methods

### General Considerations

Commercial reagents were used without further purification. ^1^H NMR, ^13^C NMR spectra were recorded on a Bruker DMX400 NMR spectrometer (Billerica, MA, USA).

Powder X-ray diffraction (PXRD) data were collected on a Rigaku Ultimate-IV X-ray diffractometer (Akishima, Japan) operating at 40 kV/30 mA using the Cu K*α* line (*λ* = 1.5418 Å). Data were measured over the range of 5–45° in 5°/min steps over 8 min.

Vapochromic experiments. An open 4 mL vial containing 25 mg of TCBQ was placed in a sealed 20 mL vial containing 1 mL of each vapor amine. TCBQ powders were exposed under saturated vapor pressure in the closed vessel at room temperature. Clear color changes were observed over time.

Preparation of test papers. TCBQ (100 mg) was dissolved in 10 mL of CH_2_Cl_2_, and then 1 mL of the prepared solution was slowly dropped onto the filter paper with the size of 1 cm × 3 cm, repeating the above procedure three times before drying the test papers at room temperature for 2 h.

## 4. Conclusions

In summary, we demonstrate that the gas–solid reaction that occurs between volatile amines and TCBQ can cause significant color changes. Interestingly, the various disubstituted products of TCBQ obtained by its exposure to different vapor amines were visually distinct in color in a time-dependent manner, and overall colorizing rates were in the order of *n*-propylamine > dimethylamine > *n*-butylamine > ethylamine > *n*-amylamine > *n*-hexylamine. Moreover, TCBQ can be easily fabricated into simple and flexible rapid test strips for detecting and distinguishing *n*-propylamine from ethylamine, *n*-butyamine, *n*-pentamine, *n*-butyamine, and dimethylamine in less than 3 s without any equipment assistance. We believe that our results presented here shed light on the design of rapid, convenient, and visual amine sensing materials.

## Figures and Tables

**Figure 1 molecules-29-01818-f001:**
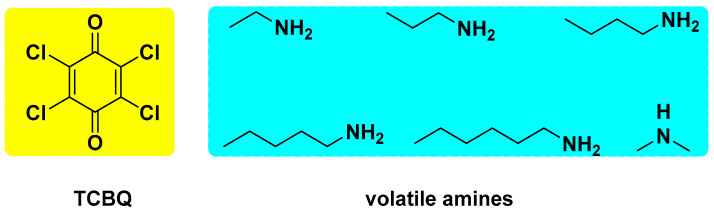
Chemical structures of TCBQ and amines.

**Figure 2 molecules-29-01818-f002:**
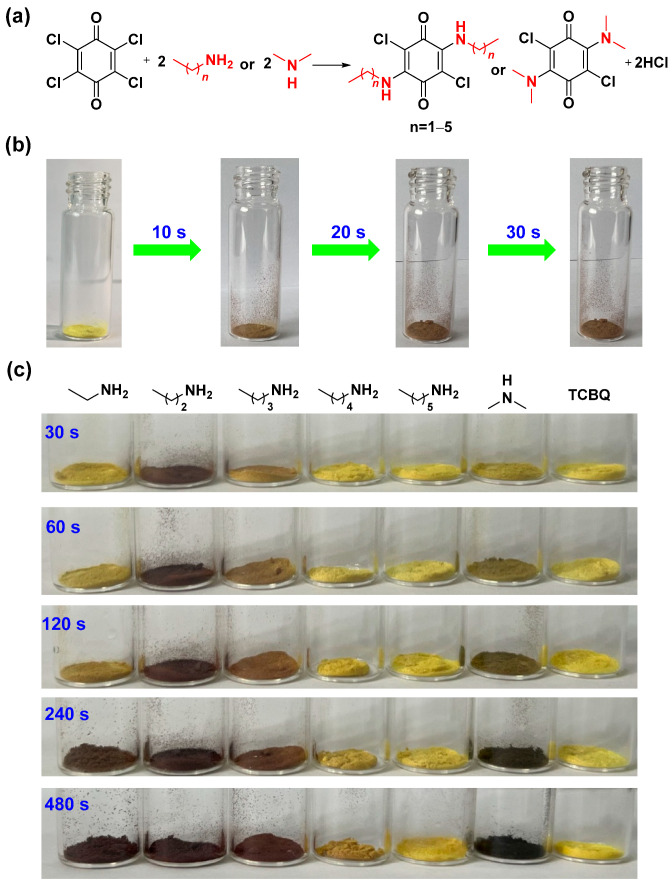
(**a**) Reaction between TCBQ and amines; (**b**) photos of TCBQ before and after exposure to *n*-propylamine vapor at different times; and (**c**) photos of TCBQ after exposure to amine vapor at different times.

**Figure 3 molecules-29-01818-f003:**
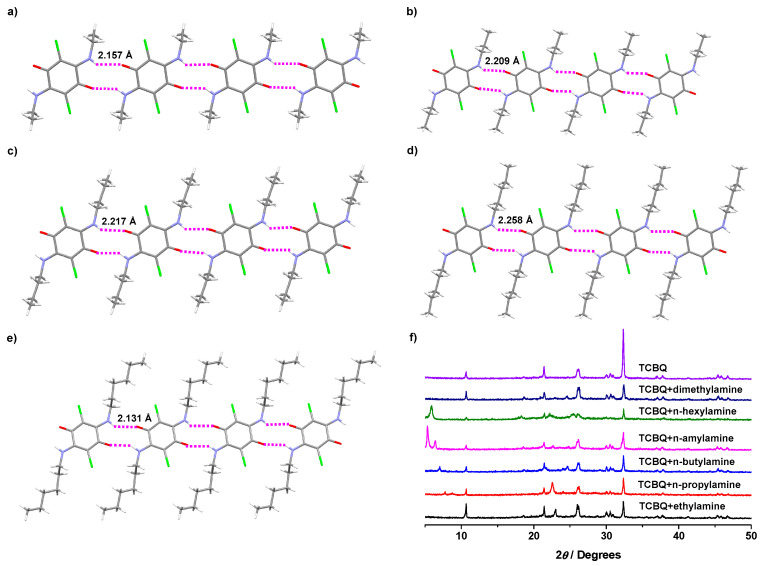
Crystal structures of disubstituted products of TCBQ with ethylamine (**a**), *n*-propylamine (**b**), *n*-butylamine (**c**), *n*-amylamine (**d**), and *n*-hexylamine (**e**), respectively (CCDC: 2345116–2345119, 2345128); (**f**) PXRD patterns of TCBQ after exposure to the amine vapor. Green color represents chlorine atoms. White color represents hydrogen atoms. Gray color represents carbon atoms. Red color represents oxygen atoms. Blue color represents nitrogen atoms. The pink dotted line represents hydrogen bond interactions.

**Figure 4 molecules-29-01818-f004:**
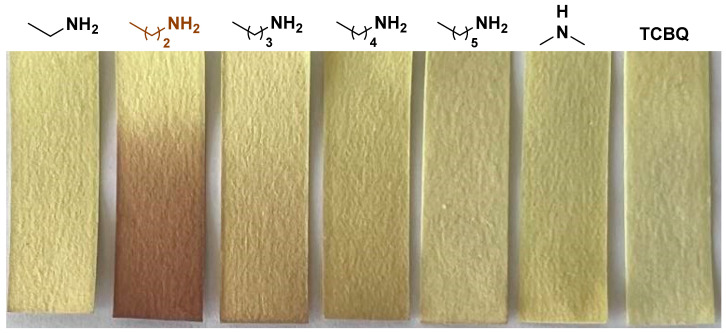
Photos of TCBQ-loaded test strips after exposure to different saturated amine vapors for 3 s.

## Data Availability

The original contributions presented in the study are included in the article/Appendix A, further inquiries can be directed to the corresponding authors.

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
