# Peer review of "Rapid and Visual Detection of Volatile Amines Based on Their Gas–Solid Reaction with Tetrachloro-p-Benzoquinone"

_molecules, 2024, doi:10.3390/molecules29081818_

Round 1

Reviewer 1 Report

Comments and Suggestions for Authors

The article presents a very interesting observation of the reaction between TCBQ and amines occurring at the solid/gas interphase. The authors did a good job in the identification of the products, including single-crystal X-ray structural analysis.  The only question here is what was the other product of this nucleophilic substitution. More specifically, if chlorines were substituted, what was the counter-ion for chloride anions which were released? The authors should provide a balanced chemical equation for their process.

 Most unexpected, however, is a drastic difference in the effects of different amines on  TCBQ-covered paper.

The first question here is about reproducibility. In other words, how many completely independent experiments (with newly prepared strips and different samples of amines) were done?

The second question is why only part of the strip changes the color? And does the position of the strips in the picture correspond to those in the vials (since the vial with a strip was placed in the vial with amine, the vapors come into contact with the strip starting from the top).   

The third question is how the time of the exposure of the strips was controlled?

Once these questions are addressed, the article can be published.

Comments on the Quality of English Language

English is fine, rather minor edits are required

Author Response

Apr 15, 2024

Dear editor:

               We are very grateful to you and the reviewers for the comments on our manuscript "Rapid and visual detection of volatile amines based on the gas-solid reaction of tetrachloro-p-benzoquinone with amines" (molecules-2969961), which are helpful for us to improve our manuscript.

According to the comments, the manuscript and SI have been revised, and the detailed changes and responses are as follows:

Reviewer 1:

  1. According to the suggestion, two molecules HCl have been added in the chemical equation. (see Figure 2a)
  2. The test paper for the detection of different amine samples were done at least three times, and the same results were obtained.
  3. Yes, the position of the strips in the picture corresponds to those in the vials.
  4. We start the timing by putting the test paper into the vial, and when we see a significant color change, we stop the timing and immediately take out the test paper.

Thank you and the reviewers for the valuable comments and suggestions again.

Sincerely yours,

Fei Zeng

Hunan University of Science and Engineering

Yongzhou, 425199

Apr 15, 2024

Dear editor:

               We are very grateful to you and the reviewers for the comments on our manuscript "Rapid and visual detection of volatile amines based on the gas-solid reaction of tetrachloro-p-benzoquinone with amines" (molecules-2969961), which are helpful for us to improve our manuscript.

According to the comments, the manuscript and SI have been revised, and the detailed changes and responses are as follows:

Reviewer 1:

  1. According to the suggestion, two molecules HCl have been added in the chemical equation. (see Figure 2a)
  2. The test paper for the detection of different amine samples were done at least three times, and the same results were obtained.
  3. Yes, the position of the strips in the picture corresponds to those in the vials.
  4. We start the timing by putting the test paper into the vial, and when we see a significant color change, we stop the timing and immediately take out the test paper.

Thank you and the reviewers for the valuable comments and suggestions again.

Sincerely yours,

Fei Zeng

Hunan University of Science and Engineering

Yongzhou, 425199

China

Tel: 86-15869977707

China

Tel: 86-15869977707

Reviewer 2 Report

Comments and Suggestions for Authors

The manuscript titled "Rapid and visual detection of volatile amines based on the gas-solid reaction of tetrachloro-p-benzoquinone with amines" by Sun et al. presents a new method for the detection of volatile amines using tetrachloro-p-benzoquinone (TCBQ) as a sensor. The authors demonstrate the gas-solid reaction between TCBQ and various volatile amines, leading to visually distinct color changes. They also fabricate test strips for rapid and equipment-free detection of specific amines. Overall, the manuscript is well-written, and the research is novel and significant. However, a few areas require minor revision for clarification and improvement before the manuscript can be published.

Specific Comments:

1. The introduction provides a good background on the importance of detecting volatile amines and the limitations of existing methods. However, it would be helpful to include some discussion on the significance of the gas-solid reaction approach and how it overcomes the drawbacks of previous techniques.

2. In the experimental section, more details regarding the fabrication of the test strips should be provided. The materials and methods for preparing the test strips should be clearly described, including any additional components or modifications to the TCBQ. 

3. The results and discussion section provides comprehensive information on the gas-solid reactions, color changes, and detection performance of TCBQ with different volatile amines. However, the discussion could be further expanded to provide insights into the underlying mechanisms of the gas-solid reactions and the factors influencing the color changes. Additionally, the authors should discuss the specificity and selectivity of the TCBQ-based test strips in distinguishing n-propylamine from other volatile amines.

4. Minor language issues should be addressed throughout the manuscript. For example, the sentence “traditional amines detection methods are required bulky equipment size, expensive cost and complexity in measurement” in the abstract is grammatically incorrect.

Overall, the manuscript presents a promising approach for the rapid detection of volatile amines using TCBQ. With some revisions and clarifications, this work has the potential for publication.

Comments on the Quality of English Language

Minor language issues should be addressed throughout the manuscript. 

Author Response

Apr 15, 2024

Dear editor:

               We are very grateful to you and the reviewers for the comments on our manuscript "Rapid and visual detection of volatile amines based on the gas-solid reaction of tetrachloro-p-benzoquinone with amines" (molecules-2969961), which are helpful for us to improve our manuscript.

According to the comments, the manuscript and SI have been revised, and the detailed changes and responses are as follows:

Reviewer 2:

1.      According to the suggestion, the discussion on the significance of the gas-solid reaction has been added in the manuscript. (page 2)

  1. According to the suggestion, the detail for the fabrication of the test strips has been added in the manuscript. (Preparation of test papers part)

3.      According to the suggestion, the mechanisms of the gas-solid reactions and the factors influencing the color changes time of the color change was added in the manuscript (page 2). However, it is still unclear why propylamine and TCBQ have the fastest reaction times.4.      According to the suggestion, “traditional amines detection methods are required bulky equipment size, expensive cost and complexity in measurement” has been changed to “traditional amines detection methods require bulky equipment size, expensive cost and complexity in measurement”.

Thank you and the reviewers for the valuable comments and suggestions again.

Sincerely yours,

Fei Zeng

Hunan University of Science and Engineering

Yongzhou, 425199

China

Tel: 86-15869977707
